# Forced Apart: Discovering Disentangled Representations Without Exhaustive Labels

## Abstract

Learning a better representation with neural networks is a challenging problem, which has been tackled from different perspectives in the past few years. In this work, we focus on learning a representation that would be useful in a clustering task. We introduce two novel loss components that substantially improve the quality of produced clusters, are simple to apply to arbitrary models and cost functions, and do not require a complicated training procedure. We perform an extensive set of experiments, supervised and unsupervised, and evaluate the proposed loss components on two most common types of models, Recurrent Neural Networks and Convolutional Neural Networks, showing that the approach we propose consistently improves the quality of KMeans clustering in terms of mutual information scores and outperforms previously proposed methods.

## 1 Introduction

Representation learning is an important part of deep learning research, and the ability of deep neural networks to transform the input data into a space that is more suitable to the target task is one of the key reasons for their success. Consider the case of binary classification with a neural network with sigmoid activation function on the last layer, where a network transforms the input data $x \in \mathbb{R}^n$ into a space $\mathbb{R}$ where two classes are linearly separable by applying a sequence of non-linear transformations

$$f(x) : \mathbb{R}^n \to \mathbb{R}^{k_1} \to \mathbb{R}^{k_2} \to \cdots \to \mathbb{R}^{k_j} \to \mathbb{R}$$

Note that all representations, learned by the network in the sequence of transformations $\mathbb{R}^i \to \mathbb{R}^j$, are devoted to one goal: binary classification. The learned intermediate representations can easily be used in tasks similar to the binary classification, but using them in a different task may be problematic.

Consider the case of multivariate time series classification with an RNN model, depicted in Figure 1 with a sigmoid activation function in the last $FC_2$ layer and a ReLU activation function in the layer $FC_1$. Note that ReLU activation produces non-negative vectors. During a regular training procedure with binary cross-entropy loss, the model will learn weights that produce two patterns of activation of the layer $FC_1$: roughly orthogonal vectors for the samples that belong to different classes, and roughly parallel vectors for the samples that belong to the same class. Indeed, the value of the output scalar is the result of taking the dot product between the weights $\mathbf{w}$ of the final layer $FC_2$ (a single vector in this case) and the output $\mathbf{h}$ of the penultimate hidden layer $FC_1$. Via the geometric interpretation of the dot product, this value is highest when the cosine between the vectors 1, and minimized when the cosine is $-1$. However, since the penultimate layer has the ReLU activation, the vectors cannot point in opposite directions, therefore, they must be orthogonal.

$$\max \boldsymbol{w}^T \boldsymbol{h} = \max \|\boldsymbol{w}\| \|\boldsymbol{h}\| \cos\theta \Rightarrow \theta = 0 \tag{1}$$

$$\min \boldsymbol{w}^T \boldsymbol{h}, \boldsymbol{h} \geq 0 = \min \|\boldsymbol{w}\| \|\boldsymbol{h}\| \cos\theta \Rightarrow \theta = \frac{\pi}{2} \tag{2}$$

$$\boldsymbol{h}_i \| \boldsymbol{h}_j, \text{if } y_i = y_j \tag{3}$$

$$\boldsymbol{h}_i \perp \boldsymbol{h}_i, \text{if } y_i \neq y_j \tag{4}$$

where $y_i$ is the corresponding binary label for hidden state $\boldsymbol{h_i}$.

In this work, we focus on learning a better representation of the input that could be used in downstream tasks such as clustering. Specifically, we are interested in learning the representation that would enable clustering by virtue of revealing its latent structure, while using the limited information provided by the binary classification task. In order to force the network to learn such diverged representations, we propose two novel loss components that can be applied to an arbitrary cost function and work in both weakly-supervised and unsupervised settings. We evaluate the proposed loss components empirically on two most common types of models, Recurrent Neural Networks (RNN) and Convolutional Neural Networks (CNN) and different types of input data (time series, images, texts). Our approach shows consistent improvement of the quality of KMeans clustering in terms of mutual information scores, outperforming previous methods.

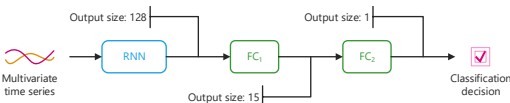

Figure 1: An RNN model with two fully-connected layers for binary classification of time series

## 2 RELATED WORK

In the past few years, a substantial amount of work has been dedicated to learning a better representation of the input data that can be either used in downstream tasks, such as KMeans clustering, or to improve generalizability or performance of the model. In general, these works can be divided into three categories: (1) approaches that introduce a new loss component that can be easily applied to an arbitrary cost function (discriminative models), (2) approaches that require a complicated or cumbersome training procedure (discriminative models), and (3) probabilistic generative and/or adversarial models.

Approaches from the first group propose new loss components that can be applied in a straightforward manner to an arbitrary cost function, supervised or unsupervised. Cheung et al. (2014) proposed a cross-covariance penalty (`XCov`) to force the network to produce representations with disentangled factors. The proposed penalty is, essentially, cross-covariance between the predicted labels and the activations of samples in a batch. Their experiments showed that the network can produce a representation, with components that are responsible to different characteristics of the input data. For example, in case of the MNIST dataset, there was a class-invariant factor that was responsible for the style of the digit, and in case of the Toronto Faces Dataset (Susskind et al., 2010), there was a factor responsible for the subject's identity. Similarly, but with a different goal in mind, Cogswell et al. (2015) proposed a new regularizer (`DeCov`), that minimizes cross-covariance of hidden activations, leading to non-redundant representations and, consequently, less overfitting and better generalization. `DeCov` loss is trying to minimize the Frobenius norm of the covariance

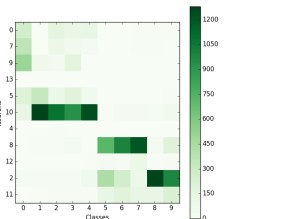

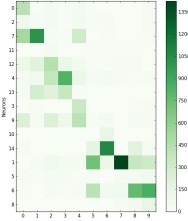

(a) Without the proposed loss component

(b) With the proposed loss component $\mathcal{L}_{single}$

Figure 2: Number of samples for which the neurons on the $y$ axis were active the most in a binary classification task on MNIST strokes sequences dataset. The classes 0-4 have the label 0, and the classes 5-9 have the label 1. See the subsection 4.1 and subsection 5.2 for details.

matrix between all pairs of activations in the given layer. The authors' experiments showed that the proposed loss significantly reduced overfitting and led to a better classification performance on a variety of datasets.

The second group of methods requires a modification of the standard training procedure with back-propagation and stochastic gradient descent optimizers. Liao et al. (2016) proposed a method to learn parsimonious representations. Essentially, the proposed algorithm iteratively calculates cluster centroids, which are updated every $M$ iterations and used in the cost function. The authors' experiments showed that such algorithm leads to a better generalization and a higher test performance of the model in case of supervised learning, as well as unsupervised and even zero-shot learning. Similarly, Xie et al. (2016) proposed an iterative algorithm that first calculates soft cluster assignments, then updates the weights of the network and cluster centroids. This process is repeated until convergence. In contrast to Liao et al. (2016), the authors specifically focused on the task of learning better representations for clustering, and showed that the proposed algorithm gives a significant improvement in clustering accuracy.

Finally, a new group of recently emerged methods focus on disentangling the factors of variation (e.g., style and class). Kingma et al. (2014) proposed deep generative models for semi-supervised learning and showed that is possible to generate samples from the target class with variations in style, and vice versa. Makhzani et al. (2015) proposed a new approach, called adversarial autoencoder (AAE) and performed a variety of experiments, including semi-supervised and unsupervised clustering, achieving impressive results on MNIST (LeCun et al., 1998) and Street View House Numbers (Netzer et al., 2011) datasets. However, since this methods includes adversarial networks, the training of such systems is rather cumbersome. For example, in the semi-supervised autoencoders experiments, the training of the system consisted of three different phases: a reconstruction phase, a regularization phase, and a semi-supervised classification phase, where the regularization phase itself consists of two sub-phases of updating discriminator and generator respectively. Finally, (Mathieu et al., 2016) proposed a conditional generative model that is a combination of Variational Autoencoder (Kingma & Welling, 2013) and Generative Adversarial Networks (Goodfellow et al., 2014) for disentangling factors of variations.

Our proposed loss components belong to the first group and, in contrast to the other methods do not require a complicated training procedure, can easily be used with any cost function, and work in both weakly-supervised and unsupervised settings.

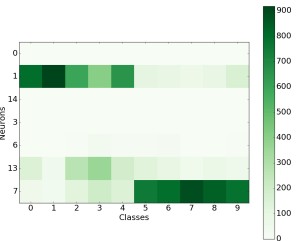
(a) Without the proposed loss component

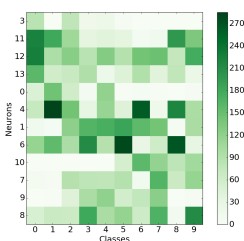
(b) With the proposed loss component $\mathcal{L}_{\text{multi}}$

Figure 3: Number of samples for which the neurons on the $y$ axis were active the most in a binary classification task on the CIFAR-10 dataset. See the subsection 4.2 for details.

## 3 THE PROPOSED METHOD

Inspired by Equation 1 and the work of Cheung et al. (2014) and Cogswell et al. (2015), we propose two novel loss components, which despite their simplicity, significantly improve the quality of the clustering over the representations produced by the model. The first loss component $\mathcal{L}_{\text{single}}$ works on a single layer and does not affect the other layers in the network, which may be a desirable behaviour in some cases. The second loss component $\mathcal{L}_{\text{multi}}$ affects the entire network behind the target layer

and forces it to produce disentangled representations in more complex and deep networks in which the first loss may not give the desired improvements.

## 3.1 SINGLE LAYER LOSS

Consider the model in Figure 1. The layer $FC_2$ has output size of 1 and produces a binary classification decision. The output of the layer $FC_1$ is used to perform KMeans clustering. Recall from the example in the introduction that we want to force the model to produce divergent representations for the samples that belong to the same class, but are in fact substantively different from each other. One way to do it would be to force the *rows* of the weight matrix $W_{FC_1}$ of the $FC_1$ layer to be different from each other, leading to different patterns of activations in the output of the $FC_1$ layer.

Formally, it can be expressed as follows:

$$\mathcal{L}_{\text{single}} = \sum_{i=1}^{k} \sum_{j=i+1}^{k} f_l(d_i, d_j) + f_l(d_j, d_i) \tag{5}$$

where $d_k$ are normalized weights of the row $k$ of the weights matrix $W$ of the given layer:

$$d_k = \text{softmax}(W[k]) \tag{6}$$

and $f_l(d_i, d_j)$ is a component of the loss between the rows $i$ and $j$:

$$f_l(x_i, x_j) = \max(0, m - D_{\text{KL}}(x_i || x_j)) \tag{7}$$

where $m$ is a hyperparameter that defines the desired margin of the loss component and $D_{\text{KL}}(d_i || d_j)$ is the Kullback–Leibler divergence[1] between the probability distributions $d_i$ and $d_j$.

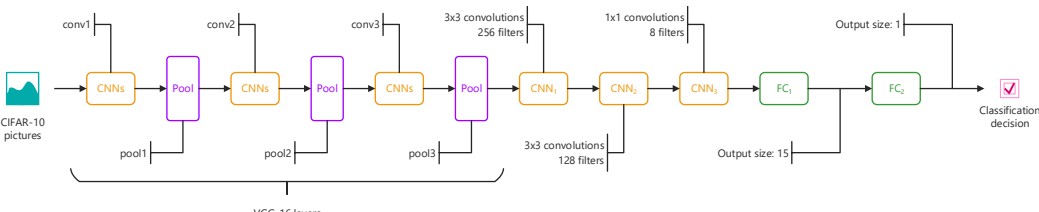

Figure 4: A CNN model used in the CIFAR-10 experiments

## 3.2 MULTILAYER LOSS

Note that the loss component $\mathcal{L}_{\text{single}}$ affects only the weights of the specific layer, as it operates not on the outputs of the layer but directly on its weights, similar to, for example, $\ell_2$ regularization. Therefore, this loss component may help to learn a better representation only if the input to the target layer still contains the information about latent characteristics of the input data. This might be the case in simple shallow networks, but in case of very deep complex networks the input data is non-linearly transformed so many times that only the information that is needed for binary classification left, and all the remaining latent characteristics of the input data were lost as not important for binary classification (see the Figure 3a). Indeed, as we can see from the experiments in Section 4, the loss component described above substantially improves the quality of clustering in a simple baseline case. However, in the case of a more complex model, this improvement is much less impressive. Therefore, we also propose a loss component that can influence not only one specific layer, but all layers before it, in order to force the network to produce a better representation.

Recall again that we want to force the model to produce disentangled representations of the input data. Namely, that these representations should be sufficiently different from each other even if two

---

[1]Note that the proposed framework does not limit the choice of divergence measure between the two distributions, for example, the Jensen-Shannon divergence can be used, etc.

samples have the same label. We propose the following loss component in order to produce such properties:

$$\mathcal{L}_{\text{multi}} = \frac{1}{N_s^2} \sum_{i=1}^{N} \sum_{j=1}^{N} \begin{cases} f_l(h_i^s, h_j^s) + f_l(h_j^s, h_i^s) & y_i = y_j \\ 0 & y_i \neq y_j \end{cases} \tag{8}$$

where $h_k^s$ is a normalized output of the target layer $h$ for the sample $k$:

$$h_k^s = \text{softmax}(h_k) \tag{9}$$

$y_k$ is its the ground truth label, $N$ is the number of samples in the batch, $N_s$ is number of samples that have the same label, and $f_l(h_i, h_j)$ is the function defined in Equation 7. Note that this loss component $\mathcal{L}_{\text{multi}}$ works on the outputs of the target layer, and therefore, it affects the whole network behind the layer on which it is applied, overcoming the local properties of the $\mathcal{L}_{\text{single}}$ loss.

### 3.3 Unsupervised learning

Although our main focus in the presented experiments is on a binary classification task, both of our proposed loss components can be used in unsupervised learning as well. The loss component $\mathcal{L}_{\text{single}}$ does not require any labels so it can be used without modifications. The loss component $\mathcal{L}_{\text{multi}}$ can be applied to unlabeled data by just taking the summations without consideration of labels of the samples as follows:

$$\mathcal{L}_{\text{multi}_2} = \frac{1}{N^2} \sum_{i=1}^{N} \sum_{j=1}^{N} f_l(h_i^s, h_j^s) + f_l(h_j^s, h_i^s) \tag{10}$$

For example, as autoencoder models are a common choice to learn representations to use in a downstream task, the proposed loss components can be easily applied to its cost function as follows:

$$\mathcal{L}_{ae} = (1 - \alpha) * \frac{1}{N} \sum_{i=1}^{N} ||X_i - \hat{X}_i||^2 + \alpha * \mathcal{L}_{\text{multi}} \tag{11}$$

where the first part is a standard reconstruction cost for autoencoder, the second is the proposed loss component, and $\alpha$ is a hyperparameter reflecting how much importance is given to it.

### 3.4 The margin hyperparameter $m$

One important choice to be made while using the proposed loss components is the value of the margin hyperparameter $m$. A larger value of $m$ corresponds to a larger margin between the rows of the weights matrix in case of $\mathcal{L}_{\text{single}}$ and a larger margin between the activations of the target layer in case of $\mathcal{L}_{\text{multi}}$. The smaller the value of $m$, the less influence the proposed loss components have.

In our experiments, we found that the proposed loss component $\mathcal{L}_{\text{single}}$ is relatively stable with respect to the choice of $m$, and generally performs better with larger values (in the range 5-10). In case of the loss component $\mathcal{L}_{\text{multi}}$, we found that even a small value of the margin $m$ (0.1 - 1) disentangles the learned representations better and consequently leads to substantial improvements in the AMI score.

In all of the reported experiments, we found that the proposed loss component with a reasonably chosen $m$ does not hurt the model's performance in the classification task.

## 4 Experiments

We performed an extensive set of experiments that covers the two most commonly used in modern research: Recurrent Neural Networks and Convolutional Neural Networks, as well as entirely different modalities of the input data: time series, images, and texts.

In all experiments, we used an RNN or an CNN model without any additional loss components as the baseline and compared our proposed loss components $\mathcal{L}_{\text{single}}$ and $\mathcal{L}_{\text{multi}}$ with the `DeCov` regularizer (Cogswell et al., 2015) and `XCov` penalty (Cheung et al., 2014), as those works are most

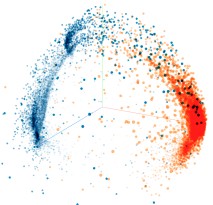
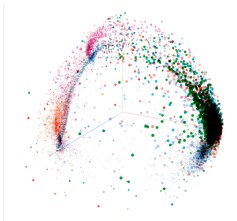

(a) Without the proposed loss component, colored by binary labels

(b) Without the proposed loss component, colored by classes

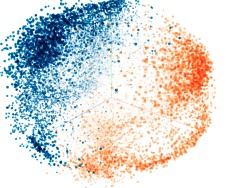
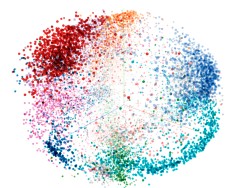

(c) With the proposed loss component $\mathcal{L}_{\text{multi}}$, colored by binary labels

(d) With the proposed loss component $\mathcal{L}_{\text{multi}}$, colored by classes

Figure 5: PCA visualizations of the learned representations on the MNIST strokes sequences dataset. See the subsection 5.2 for details.

similar to ours. After the model were trained on the binary classification task, we use the output of the penultimate layer to perform a KMeans clustering.

We implemented the models used in all experiments with TensorFlow (Abadi et al., 2016) and used Adam optimizer (Kingma & Ba, 2014) to train the them.

### 4.1 MNIST STROKES SEQUENCES EXPERIMENTS

We performed experiments on the MNIST strokes sequences dataset de Jong (2016)[2] to evaluate the proposed loss components the in case of an RNN model and time series data. This dataset contains pen strokes, automatically generated from the original MNIST dataset LeCun et al. (1998). Although the generated sequences do not always reflect a choice a human would made in order to write a digit, the strokes are consistent across the dataset.

For this experiment, we split the examples into two groups: samples belonging to the classes from 0 to 4 were assigned to the first group, and samples belonging to the classes from 5 to 9 were assigned to the second group. The model is trained to predict the *group* of a given sample and does not have any access to the underlying classes.

We used the model depicted in Figure 1 for this experiment. After the models were trained on the binary classification task, we used the output of the penultimate layer $FC_2$ to perform the KMeans clustering and evaluated the quality of the produced clustering using the original class labels as ground truth assignments.

**Autoencoder experiments** In order to investigate the influence of the proposed loss components in the autoencoder settings, we applied them to an autoencoder model that reconstructs the input sequences from the MNIST strokes sequences dataset. We did not use any label information during this experiments, and used the representation from the intermediate layer of the autoencoder to perform KMeans clustering.

### 4.2 CIFAR-10 EXPERIMENTS

In order to evaluate the proposed loss components on a different type of model and data, we performed experimented with the CIFAR-10 dataset Krizhevsky & Hinton (2009) using an CNN model.

---

[2]https://github.com/edwin-de-jong/mnist-digits-stroke-sequence-data

As in the MNIST strokes sequences experiments, we split the examples in two groups: samples belonging to the classes "airplan", "automobile", "bird", "cat", and "deer" were assigned to the first group, and samples belonging to the classes "dog", "frog", "horse", "ship", "truck" were assigned to the second group. Note that this assignment is quite arbitrary as it simply reflects the order of the labels of the classes in the dataset (namely, the labels 0-4 for the first group and the labels 4-9 for the second group). All groups contain rather different types of objects, both natural and human-made.

For these experiments, we used a CNN model based on the VGG-16 architecture (Simonyan & Zisserman, 2014), depicted on the Figure 4. We discarded the bottom fully connected and convolutional layers as, perhaps, they are too big for this dataset. Instead, we appended three convolutional layers to the output of `pool3` layer with number of filters 256, 128 and 8 correspondingly. The first two layers use 3x3 convolutions, and the last layer uses 1x1 convolutions. After that, we pass the output through a fully-connected layer of size 15 ($FC_1$), which produces the representations used in clustering, and a fully connected layer of size 1 ($FC_2$) with the sigmoid activation function to produce a binary classification decision.

### 4.3 TEXT CLASSIFICATION EXPERIMENTS

Finally, to prove a wide generalizability of the proposed loss components, we performed text classification experiments using an RNN model again, but on an entirely different type of data. Namely, we used the DBPedia ontology dataset dataset (Zhang et al., 2015), which contains titles and abstract of Wikipedia articles labeled by 14 ontology classes.

Again, we split the samples into two groups and trained the model on the binary classification task. Classes "Company", "EducationalInstitution", "Artist", "Athlete", "OfficeHolder", "MeanOfTransportation", "Building" belong to the first group, and the classes "NaturalPlace", "Village", "Animal", "Plant", "Album", "Film", "WrittenWork" belong to the second group. As in subsection 4.1, we used the model depicted on Figure 1.

### 4.4 IMPLEMENTATION DETAILS

Despite the fact the proposed loss components can be directly implemented using two nested `for` loops, such implementation will not be computationally efficient, as it will lead to a big computational graph operating on separate vectors without using full advantages of highly optimized parallel matrix computations on GPU. Therefore, it is desirable to have an efficient implementation that can use full advantage of modern GPUs. We have developed such an efficient implementation that significantly accelerates the computation of the loss component in return for a higher memory consumption by creating two matrices that contain all combinations of $d_i$ and $d_j$ from the summations in the Equation 5 and performing the operations to calculate the loss on them. We have made our implementation for TensorFlow (Abadi et al., 2016) publicly available on GitHub[3] alongside with aforementioned models from the subsection 4.1 and the subsection 4.2.

It is worth noting that since the loss component $\mathcal{L}_{\text{single}}$ operates directly on the weights of the target layer, its computational complexity does not depend on the size of the batch. Instead, it depends on the size of that layer. In contrast, the $\mathcal{L}_{\text{multi}}$ operates on the activations of the target layer on all samples in the batch, and its computational complexity depends on the number of samples in the batch. In practice, using the implementation described above, we were able to train models with batch size of 512 and higher without exhausting the GPU's memory.

## 5 RESULTS AND DISCUSSION

### 5.1 QUANTITATIVE ANALYSIS

We report the average of the Adjusted Mutual Information ($AMI_{\text{max}}$) and Normalized Mutual Information ($NMI_{\text{sqrt}}$) scores (Vinh et al., 2010) across three runs in Table 1. On the simplest MNIST strokes sequences dataset $\mathcal{L}_{\text{single}}$ outperforms all other methods, whereas on more challenging and complex datasets $\mathcal{L}_{\text{milti}}$ works the best, probably due to its ability to influence the learned repre-

---

[3]`http://github.com/placeholder/`

Table 1: Adjusted Mutual Information (AMI) and Normalized Mutual Information (NMI) scores for KMeans clustering on different datasets

| Model | MNIST Test set | | MNIST Autoencoder Test set | | CIFAR-10 Validation set | | CIFAR-10 Test set | | DBPedia Validation set | | DBPedia Test set | |
|---|---|---|---|---|---|---|---|---|---|---|---|---|
| | AMI | NMI | AMI | NMI | AMI | NMI | AMI | NMI | AMI | NMI | AMI | NMI |
| Baseline | 0.467 | 0.477 | 0.454 | 0.460 | 0.198 | 0.204 | 0.194 | 0.199 | 0.348 | 0.361 | 0.349 | 0.362 |
| DeCov | 0.287 | 0.313 | 0.443 | 0.450 | 0.175 | 0.191 | 0.176 | 0.191 | 0.271 | 0.322 | 0.27 | 0.323 |
| Xcov | 0.525 | 0.547 | 0.414 | 0.420 | 0.320 | 0.327 | 0.321 | 0.326 | 0.379 | 0.391 | 0.379 | 0.393 |
| L_single | **0.544** | **0.553** | 0.457 | 0.463 | 0.238 | 0.245 | 0.239 | 0.246 | 0.455 | 0.464 | 0.451 | 0.461 |
| L_multi | 0.502 | 0.523 | **0.463** | **0.470** | **0.376** | **0.384** | **0.376** | **0.385** | **0.520** | **0.523** | **0.529** | **0.533** |

sentations on all layers of the network behind the target layer. The proposed loss components also improves the quality of clustering in the autoencoder settings, although the gain is marginal.

It is also important to note that in all our experiments accuracy of models was not affected in a harmful way when we applied the proposed loss components, or the effect was negligible (less than 0.5%).

## 5.2 QUALITATIVE ANALYSIS

To examine the influence of the proposed loss components to the activations of the network, we plot the number of samples, belonging to different underlying classes on the $x$ axis for which the neurons on the $y$ axis were active the most in the binary classification task on Figure 2 and Figure 3 for on MNIST strokes sequences and CIFAR-10 datasets correspondingly. As we can see from these figures, during a regular training with the binary classification objective, without the proposed loss component the models tend to learn representations that is specific to the target binary label, even though the samples within one group come from different classes. The model learns to use mostly just two neurons to discriminate between the target groups and hardly uses the rest of the neurons in the layer. We observe this behaviour across different types of models and datasets: an RNN model applied to a timeseries dataset and an CNN model applied to an image classification dataset behave in the exactly the same way. Both proposed loss components $\mathcal{L}_{single}$ and $\mathcal{L}_{multi}$ force the model to produce diverged representations, and we can see how it changes the patterns of activations in the target layer. It is easy to observe in Figure 2b that the patterns of activations learned by the networks roughly correspond to underlying classes, despite the fact that the network did not have access to them during the training. This pattern is not as easy to see in case of CIFAR-10 dataset (see the Figure 3b), but we can observe that the proposed loss component nevertheless forced the network to activate different neurons for different classes, leading to a better AMI score on the clustering task.

In order to further investigate the representations learned by the model, we visualized the representations of samples from the MNIST strokes sequences dataset in Figure 5 using TensorBoard. Figure 5a and Figure 5b in the top row depict the representations learned by the baseline model, colored according to the binary label and the underlying classes, respectively. Figure 5c and Figure 5d in the bottom row depict the representations of the same samples, learned by the model with the $\mathcal{L}_{multi}$ loss component, colored in the same way. It is easy to see that the $\mathcal{L}_{multi}$ indeed forced the model to learn disentangled representations of the input data. Note how the baseline model learned dense clusters of objects, with samples from the same group (but different classes) compactly packed in the same area. In contrast, the model with the proposed loss component learned considerably better representations which disentangle samples belonging to different classes and placed the them more uniformly in the space.

In the real world, the number of clusters is rarely known beforehand. To systematically examine the stability of the proposed loss component, we plotted the Adjusted Mutual Information scores for the baselines methods and $\mathcal{L}_{multi}$ loss component with respect to the number of clusters in Figure 6, using the CIFAR-10 dataset. As can be seen from Figure 6, our loss component consistently outperforms the previously proposed methods regardless the number of clusters.

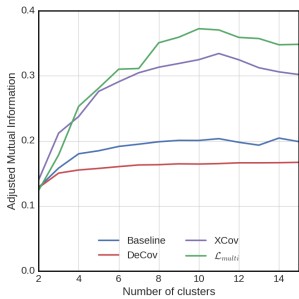

Figure 6: Number of clusters and the corresponding AMI score on the CIFAR-10 dataset

## 6 CONCLUSION

In this paper, we propose two novel loss components that substantially improve the quality of KMeans clustering, which uses representations of the input data learned by a given model. We performed a comprehensive set of experiments using two popular neural network architectures (RNNs and CNNs), and different modalities of data (image and text). Our results demonstrate that the proposed loss components consistently increase the Mutual Information scores by a significant margin, and outperform previously proposed methods. In addition, we qualitatively analyzed the representations learned by the network by visualizing the activation patterns and relative positions of the samples in the learned space, showing that the proposed loss components indeed force the network to learn diverged representations.

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
