# OpenReview forum: "Forced Apart: Discovering Disentangled Representations Without Exhaustive Labels"
_ICLR.cc/2018/Conference — Reject_

### Official Review · AnonReviewer1 · 2017-11-27
**This paper proposes two regularization terms to encourage learning disentangled representations.**

**Rating:** 5
**Confidence:** 5

**Review:**

This paper proposes two regularization terms to encourage learning disentangled representations. One term is applied to weight parameters of a layer just like weight decay. The other is applied to the activations of the target layer (e.g., the penultimate layer). The core part of both regularization terms is a compound hinge loss of which the input is the KL divergence between two softmax-normalized input arguments. Experiments demonstrate the proposed regularization terms are helpful in learning representations which significantly facilitate clustering performance.

Pros:
(1) This paper is clearly written and easy to follow.

(2) Authors proposed multiple variants of the regularization term which cover both supervised and unsupervised settings.

(3) Authors did a variety of classification experiments ranging from time serials, image and text data.

Cons:
(1) The design choice of the compound hinge loss is a bit arbitrary. KL divergence is a natural similarity measure for probability distribution. However, it seems that authors use softmax to force the weights or the activations of neural networks to be probability distributions just for the purpose of using KL divergence. Have you compared with other choices of similarity measure, e.g., cosine similarity? I think the comparison as an additional experiment would help explain the design choice of the proposed function.

(2) In the binary classification experiments, it is very strange to almost randomly group several different classes of images into the same category. I would suggest authors look into datasets where the class hierarchy is already provided, e.g., ImageNet or a combination of several fine-grained image classification datasets.

Additionally, I have the following questions:
(1) I am curious how the proposed method compares to other competitors in terms of the original classification setting, e.g., 10-class classification accuracy on CIFAR10.
(2) What will happen for the multi-layer loss if the network architecture is very large such that you can not use large batch size, e.g., less than 10?

(3) In drawing figure 2 and 3, if the nonlinear activation function is not ReLU, how would you exam the same behavior? Have you tried multi-class classification for the case “without proposed loss component” and does the similar pattern still happen or not?

Some typos:
(1) In introduction, “when the cosine between the vectors 1” should be “when the cosine between the vectors is 1”.

(2) In section 4.3, “we used the DBPedia ontology dataset dataset” should be “we used the DBPedia ontology dataset”.

I would like to hear authors’ feedback on the issues I raised.

---

> ### Author Response · Authors · 2017-12-28
> **Response to AnonReviewer1**
>
> Q. Have you compared with other choices of similarity measure, e.g., cosine similarity?
> A. We performed additional experiments on MNIST strokes sequences dataset and on the DBPedia dataset using the cosine similarity on weights, cosine similarity on activations, and orthogonality regularization on the weights.  All of these improved the quality of clusterization in terms of AMI and NMI, however, the regularizations we propose in this paper perform the best (see the results table in the general response section above).  We hypothesize that the reason for this is that the proposed method involves non-linear relationships between the change in the weights and the corresponding loss, whereas, for example, cosine similarity does not.
>
> Q. In the binary classification experiments, it is very strange to almost randomly group several different classes of images into the same category...
> A. Our goal was to look at the case where labeled classes are composed from different “types” (sub-classes) of objects. In a sense, this is a hierarchical classification, where only the labels of the first level is accessible to the network.  However, we agree that additional experiments using a proper hierarchical dataset would also be informative, and we will include them.
>
> Q. I am curious how the proposed method compares to other competitors in terms of the original classification setting.
> A. We compared the classification accuracy on the binary classification task and there was no effect (or a negligible effect) of the proposed method on the classification accuracy.
>
> Q. What will happen for the multi-layer loss if the network architecture is very large such that you can not use large batch size, e.g., less than 10?
> A: Since the batches are usually formed randomly, chances are there will be samples with the same label (this is the always the case in the batch of 3 or more) and different underlying groups. Since the training is repeated for many batches, this should not be an issue.
>
> Q. In drawing figure 2 and 3, if the nonlinear activation function is not ReLU, how would you exam the same behavior?
> A: Actually, we experimented with different activation functions (ReLU, sigmoid, tanh) and they all had the same behaviour. We will mention this in the paper.
>
> Q. Have you tried multi-class classification for the case “without proposed loss component” and does the similar pattern still happen or not?
> A. We have not tried this. We expect the same behaviour as it was general for such diverse architectures and tasks, and the binary classification is just a particular case of a multi-class classification. We will, however, perform additional experiments to verify it empirically.

---

### Official Review · AnonReviewer2 · 2017-11-27
**Needs comparison to other baselines**

**Rating:** 5
**Confidence:** 4

**Review:**

Summary
This paper proposes two regularizers that are intended to make the
representations learned in the penultimate layer of a classifier more conforming
to inherent structure in the data, rather than just the class structure enforced
by the classifier. One regularizer encourages the weights feeding into the
penultimate layer to be dissimilar and the other encourages the activations
across samples (even if they belong to the same class) to be dissimilar.

Pros
- The proposed regularizers are able to separate out the classes inherent in the
  data, even if this information is not provided through class labels. This is
validated on several datasets using visualizations as well as quantitative
metrics based on mutual information.

Cons
- It is not explained why it makes sense to first convert the weight vectors
  into probability distributions by applying the softmax function, and then
measuring distances using KL divergence between the probability distributions.
It should be explained more clearly if there is there a natural interpretation
of the weight vectors as probability distributions. Otherwise it is not obvious
why the distance between the weight vectors is measured the way it is.

- Similarly, the ReLU activations are also first converted into probability
  distributions by applying a softmax. It should be explained why the model does
this, as opposed to simply using dot products to measure similarity.

- The model is not compared to simpler alternatives such as adding an
  orthogonality regularization on the weights, i.e., computing W^TW and making
the diagonals close to 1 and all other terms 0. Similar regularizers can be
applied for activation vectors as well.

- The objective of this paper seems to be to produce representations that are
  easy to separate into clusters. This topic has a wealth of previous work. Of
particular relevance are methods such as t-SNE [1], parametric t-SNE [2], and
DEC [3]. The losses introduced in this paper are fairly straight-forward.
Therefore it would be good to compare to these baselines to show that a simple
loss function is sufficient to achieve the objective.

- Disentangling usually refers to disentangling factors of variation, for
  example, lighting, pose, and object identity which affect the appearance of a
data point. This is different from separability, which is the property of a
representation that makes the presence of clusters evident. This paper seems to
be about learning separable representations, whereas the title suggests that it
is about disentangled ones.

Quality
The design choices made in the paper (such as the choice of distance function)
is not well explained. Also, given that the modifications introduced are quite
simple, it can be improved by doing more thorough comparisons to other
baselines.

Clarity
The paper is easy to follow.

Originality
The novel aspect of the paper is the way distance is measured by converting the
weights (and activations) to probability distributions and using KL divergence
to measure distance. However, it is not explained what motivated this choice.

Significance
The objective of this model is to produce representations that are separable, which
is of general interest. However, given the wealth of previous work done in
clustering, this paper would only be impactful if it compares to other hard
baselines and shows clear advantages.

[1] van der Maaten, Laurens and Hinton, Geoffrey. Visualizing
data using t-SNE. JMLR, 2008.

[2] van der Maaten, Laurens. Learning a parametric embedding
by preserving local structure. In International Conference
on Artificial Intelligence and Statistics, 2009.

[3] Junyuan Xie, Ross Girshick, and Ali Farhadi. Unsupervised deep embedding for
clustering analysis. ICML 2016.

---

> ### Author Response · Authors · 2017-12-28
> **Response to AnonReviewer2**
>
> Q. Why is softmax applied to weight vectors and ReLU activations to convert them to  probability distributions, as opposed to, for example, using dot products or other simpler alternatives to measure similarity?
>
> A. We chose the proposed measure based on an intuition that its non-linear nature would be better suitable in this case as opposed to, for example, dot product.  In the general response section above, we report additional experiments on the MNIST strokes sequences and on DBPedia using the W^TW regularization, as well as cosine similarity on weights and cosine similarity on activations.  All of these improved the quality of clusterization in terms of AMI and NMI, however, the regularizations we propose in the paper perform the best.  Interestingly, the general approach of forcing apart weights/activation improves the quality of the clustering regardless of the particular similarity measure used.
>
> Q. The objective of this paper seems to be to produce representations that are
> easy to separate into clusters. This topic has a wealth of previous work. Of
> particular relevance are methods such as t-SNE [1], parametric t-SNE [2], and
> DEC [3].
> A. Note that the goal of the proposed method is quite different from the objective of t-SNE, which is pure dimensionality reduction. Our goal, in contrast, is to learn a representation suitable for clustering in the absence of exhaustive labels that would allow the model to learn this representation explicitly from supervision while solving a classification task.  Essentially, we are trying to address a different task, even though it also relates to recovering latent structure in the data.
>
> Q. Disentangling usually refers to disentangling factors of variation. This paper seems  to be about learning separable representations.
> A. Thank you for catching that, we were actually aware of this, and have planned to change the title to avoid the confusion.

---

### Official Review · AnonReviewer3 · 2017-11-28
**Desperately seeking neural network representations to be more useful for clustering tasks**

**Rating:** 4
**Confidence:** 4

**Review:**

The paper proposes techniques for encouraging neural network representations to be more useful for clustering tasks. The paper contains some interesting experimental results, but unfortunately lacks concise motivation and description of the method and quality of writing.

Introduction:
The introduction is supposed to present the problem and the 'chain of events' that led to this present work, but does not do that. The first paragraph contains a too length explanation that in a classification task, representations are only concerned about being helpful for this task, and not any other task. The paragraph starting with 'Consider the case...', describes in detailed some specific neural network architecture, and what will happen in this architecture during training. The main problem with this paragraph is that it does not belong in the introduction. Indeed, other parts of the introduction have no relation to this paragraph, and the first part of the text that related to this paragraph appears suddenly in Section 3. The fact that this paragraph is two thirds of the introduction text, this is very peculiar.

Furthermore, the introduction does not present the problem well:
1) What does is a better representation for a clustering task?
2) Why is that important?

Method:
There are a few problematic statements in this part:
"The first loss component L_single works on a single layer and does not affect the other layers in the network". This is not exactly true, because it affect the layer it's related to, which affect upper layers through their feedforward input or bottom layer through the backward pass.
"Recall from the example in the introduction that we want to force the model to produce divergent representations for the samples that belong to the same class, but are in fact substantively different from each other". It is not clear why this is a corollary of the example in the introduction (that should be moved to the method part).
"this loss component may help to learn a better representation only if the input to the target layer still contains the information about latent characteristics of the input data". What does this mean? The representation always contains such information, that is relevant to the task at hand...
And others. The main problem is that the work is poorly explained: starting from the task at hand, through the intuition behind the idea how to solve it.

The experiments parts contains results that show that the proposed method is superior by a substantial margin over the baseline approaches. However, the evaluation metrics and procedure are poorly explained; What are Adjusted Mutual Information (AMI) and Normalized Mutual Information  (NMI)? How are they calculated? Or at least, the mutual information between what and what are they measuring?

---

> ### Author Response · Authors · 2017-12-28
> **Response to AnonReviewer3**
>
> Most of the comments in this review seem to stem from the reviewer’s issues with presentation and writing clarity, rather than the substantive proposal in this paper.  While it seems that the other reviewers found the presentation to be clear, we will make an earnest attempt to address the concerns with the presentation flow raised in this review.
>
> Q. What are Adjusted Mutual Information (AMI) and Normalized Mutual Information  (NMI)? How are they calculated? Or at least, the mutual information between what and what are they measuring?
> A. MI-based measures we use to evaluate clustering solutions are quite standard, and we give a reference to the paper that explains them in detail.  However, we would be happy to include definitions in the camera-ready version of the paper.
>
> Q. “this loss component may help to learn a better representation only if the input to the target layer still contains the information about latent characteristics of the input data" What does this mean? The representation always contains such information, that is relevant to the task at hand…
> A. As we explain in the introduction, for example, if the target task is binary classification, the representation learned by the network may only contain the information relevant to that task. As a result, clustering the data according to other latent characteristics may be impossible, since they might not be captured in this representation.

---

### Author Response · Authors · 2017-12-28
**General response**

We thank the reviewers, particularly reviewers #1 and #2, for their detailed and constructive comments.

To reiterate, in this paper, our goal was to learn a representation suitable for clustering in the absence of exhaustive labels that would allow the model to learn this representation explicitly from supervision while solving a classification task.  We proposed two novel loss components that substantially improve the quality of produced clusters, are simple to apply to arbitrary models and cost functions, and do not require a complicated training procedure.

The main concerns raised by the reviewers related to the choice of the similarity measure in the proposed regularization method, and its relative value compared to simpler methods such as dot products or cosine similarity on weights or activations.
We performed additional experiments on MNIST strokes sequences and on DBPedia using additional baseline methods mentioned by the reviewers, including cosine similarity on weights, cosine similarity on activations, and orthogonality regularization on the weights.  Please see the results in the table below.

There are two main takeaways from these experiments:
 1. The proposed general approach of forcing apart weights/activation improves the quality of the clustering regardless of the particular similarity measure used.
 2. While in all of these experiments, the quality of clusterization in terms of AMI and NMI was improved, the regularizations we proposed in the paper consistently performed the best.

+--------------------------------------------------------------------+
|                                  | MNIST            | DBPedia          |
+                                  +----------------------------------------+
|                                  | AMI   | NMI   | AMI    | NMI   |
+--------------------------------------------------------------------+
| W^TW                      | 0.523 | 0.530 | 0.350 | 0.366 |
+--------------------------------------------------------------------+
| Weights cosine      | 0.520 | 0.527 | 0.385 | 0.412 |
+--------------------------------------------------------------------+
| Activations cosine | 0.516 | 0.550 | 0.384 | 0.449 |
+--------------------------------------------------------------------+
| Proposed                | 0.544 | 0.553 | 0.529 | 0.533 |
+--------------------------------------------------------------------+

---

### Decision · Program_Chairs · 2018-01-29
**ICLR 2018 Conference Acceptance Decision**

**Decision:**

Reject

**Comment:**

The paper proposes two regularizers for encouraging "clustered feature embeddings" (use of "disentangled" in the title is misleading). Reviewers have raised points about the lack of proper motivation and justification of the regularizers. There are also concerns on the experiments conducted to evaluate the method, including for hierarchical classification setting. Missing comparison with relevant baselines has also been pointed out as a weakness. I feel the work is not yet mature.